# Long-term effects of group rights to fisheries: Evaluating the Western Alaska Community Development Quota program

**Matthew Berman** [ORCID] *

Institute of Social and Economic Research, University of Alaska Anchorage, Anchorage, Alaska, United States of America

* matthew.berman@uaa.alaska.edu

**Data Availability Statement:** Access to the data is highly restricted by US federal law. Researchers may apply for supervised access to data by applying to the US Census Center for Economic

## Abstract

Restricting access to fisheries and other common property resources through creation of individual transferable rights has been documented to create wealth and promote conservation, but has also reduced employment and increased inequality in fishing communities. Creating group rights instead of individual rights has been suggested as an alternative strategy that could realize the benefits with diminished social cost; however, little independent evaluation of actual implementations of group rights to fisheries has occurred. The Western Alaska Community Development Quota (CDQ) program represents an example of allocation of group fishery rights to six not-for-profit organizations representing 65 small, largely Indigenous coastal communities. Using a unique data set of individual and household survey records spanning more than 25 years, we applied a difference-in-differences approach to measure changes in a variety of social and economic indicators, including Indigenous language use and educational attainment, employment, earnings, income, and poverty status, while controlling for demographic and general economic changes over the years. We found significant differences in outcomes for individuals and households in CDQ-participating communities from those residing in nearby communities ineligible for participation. Differences were especially pronounced for earnings and income. Results suggest that group rights can provide significant social benefits. The relatively small community populations provides insufficient power to determine statistically whether the benefits of the CDQ program have been increasing or diminishing over the years, or whether some communities have benefited more than others.

## Introduction

Creating individual harvest rights to fisheries and other common property resources is widely known to have increased the value of harvests while promoting resource conservation [1,2]. Although implementing individual transferable quotas (ITQs) has raised total incomes of harvesters, the increased wealth has often been accompanied by loss of jobs from fleet consolidation, increased inequality, and depopulation of fishing communities [3–7]. Creating

Studies Research Data Program. Information about where and how to apply are provided in the following document: https://manager.researchdatagov.org/RDG_User_Guide.pdf. Questions about applications may be directed to the following e-mail address: CED.FSRDC.INFO@census.gov.

**Funding:** The research received financial support from the National Science Foundation (US), awards #1216399 and #2032786. The National Science Foundation had no role in study design, data collection and analysis, decision to publish, or preparation of the manuscript.

**Competing interests:** The authors have declared that no competing interests exist.

community or group rights has been suggested as an alternative strategy that could realize the benefits with diminished social cost [7–9]. Relatively few programs have been introduced that implement group rights to fisheries or other depletable resources on a regional scale, and there has been little independent evaluation of those that do exist [10].

The Western Alaska Community Development Quota (CDQ) program represents a unique example of allocation of group harvesting rights to a significant portion of one of the world's largest industrial fisheries. Started in the 1990s, the CDQ program allocated harvest rights to six not-for-profit organizations representing 65 small, largely Indigenous coastal communities with extremely limited economic opportunities beyond commercial fishing [11,12]. Although internal review documents have suggested that the program has provided important benefits to communities [13,14], the only independent quantitative studies were conducted early in the program, and lacked sufficient data to evaluate the program's long-term effects [15,16].

Using a large data set of household survey records collected over more than 25 years by the U.S. Census Bureau, we evaluated the effects of the opportunity of communities to participate in the CDQ program on outcomes for a variety of social and economic indicators. We applied a difference-in-differences approach to measure changes in outcomes for residents of CDQ communities relative to changes in the same indicators for residents of culturally similar, adjacent communities that were ineligible to participate in the program, while controlling for demographic and general economic changes over the years. We found significant differences between CDQ and non-CDQ communities in outcomes for individuals and households, especially with regard to earnings and income. The results suggest that this particular implementation of group rights to resources has provided significant social benefits without exacerbating inequality. However, the relatively small community populations provide insufficient power to determine statistically whether the benefits of the CDQ program have been increasing or diminishing over the years, or whether some of the six organizations have been more successful than others in providing benefits to the communities they serve.

In the following sections we first provide context by summarizing the participating communities, the Bering Sea fishery, and the introduction and evolution of the CDQ program. Next, we outline the study approach, including specific hypotheses, data sources, and empirical methods. Then we discuss statistical results and their interpretation. We conclude with a summary of findings as they relate to the literature on group rights, study limitations, and suggestions for further research.

## Bering Sea communities and fisheries

Indigenous peoples have lived along the coast of the eastern Bering Sea and Aleutian Islands (BSAI) for thousands of years, relying on local fish and marine mammal stocks [17–19]. Russian fur traders initiated European colonization in 1741, brutally suppressing and enslaving Unangan residents of the Aleutian Islands to harvest resident sea otters, later shifting to fur seals when the otter population had been depleted. After the United States purchased Alaska from Russia in 1867, Americans continued fur seal servitude on the Pribilof Islands through most of the 20th Century [20]. Since the mid-20th Century, Indigenous peoples in the region have lived in small, remote communities without road or ferry access, engaged in a mixed economy combining subsistence-with wage labor and small-scale commercial fishing livelihoods in a system of household production [21–23]. The commercial fisheries focused primarily on salmon, but also included halibut and crab, and provided opportunities to continue subsistence practices essential to cultural vitality [15,24].

During the 1960s, an international fleet dominated by Japanese trawl vessels began to exploit the groundfish stocks off the coasts of Alaska. When the US declared its 200-mile

exclusive Economic Zone (EEZ) in 1976, the North Pacific Fishery Management Council (NPFMC), the government-industry regulatory body established to manage the EEZ off Alaska, began to impose annual total fishing quotas, and implemented a variety of measures to replace the international trawl fleet with a domestic industrial fishery. The program to develop a domestic fishery was highly successful. However, within 15 years, the open-access US fisheries had become overcapitalized, and the NPFMC began to limit access to manage for sustained yield while maintaining an economically viable fishing [25].

Indigenous land claims in Alaska had been settled with the 1971 Alaska Native Claims Settlement Act (ANCSA) [26] several years before the US declared its EEZ. ANCSA was shaped by disillusionment with outcomes of the reserve model historically used for colonial administration of Indigenous peoples in North America, implementing a variety of social innovations to promote economic development and self-reliance. These measures included allocation of 44 million acres of land to 12 regional for-profit Indigenous-owned corporations and their constituent village corporations, as well as a cash settlement. Despite the important gains ANCSA brought to Alaska Indigenous peoples generally, benefits were distributed unevenly. Most new jobs created by ANCSA were located in urban centers; with few exceptions, rural communities where the majority of Alaska Native people live remained impoverished [27].

Four of 12 regional corporations included BSAI communities within their boundaries. Although several of the 12 ANCSA regional corporations were able to acquire resource-rich or otherwise valuable land, the Aleutian Islands and Bering Sea coastal regions contained few marketable land-based resources. None of the four BSAI ANCSA Native corporations gained substantial assets to assist in economic development, and ANCSA did not address the rich marine resources of the Alaska EEZ. Participation in commercial BSAI fisheries was limited to small-scale local use. Local Indigenous people lacked large capital assets and logistical expertise that would be required to compete with the established industrial fleet.

## The CDQ program

By the early 1990s, when the US Congress began to consider rationalizing the BSAI fisheries [25], local residents had become mere observers to the extraction of the wealth of the sea from the region by a large industrial fishery that generated few if any benefits to local communities. The proposed legislation to appropriate the commons by awarding catch quota shares to the existing industrial fishing fleet alarmed community leaders, as it would mean that impoverished local communities would permanently lose the ability to develop commercial marine-based livelihoods.

In 1992, the US Congress responded to the concerns from Alaska communities by enacting a program to provide western Alaska communities an opportunity to participate in the BSAI offshore commercial fisheries [28]. The CDQ program represented a truly novel approach to dividing up rights to a fishery commons, shaped in large part by community activists from the region [29]. The program carved out a portion of the total fishing quota for each of the separately managed groundfish and crab fisheries—generally about 10 percent—to non-governmental organizations representing coalitions of the 65 communities located directly on or within 50 miles (80 km) of the Bering Sea coast (Fig 1). Although ten percent of a fishery divided among 65 communities may seem small, the enormous quantity and value of BSAI harvests represented a potentially significant increment to the cash economies of these small impoverished places. Program rules encourage communities to form consortia to receive and manage quota shares; consequently communities self-organized into six regional organizations representing between one and 20 communities each. Initially, CDQs were implemented only for pollock (*Gadus chalcogrammus*), but in 1995 the program was expanded to halibut

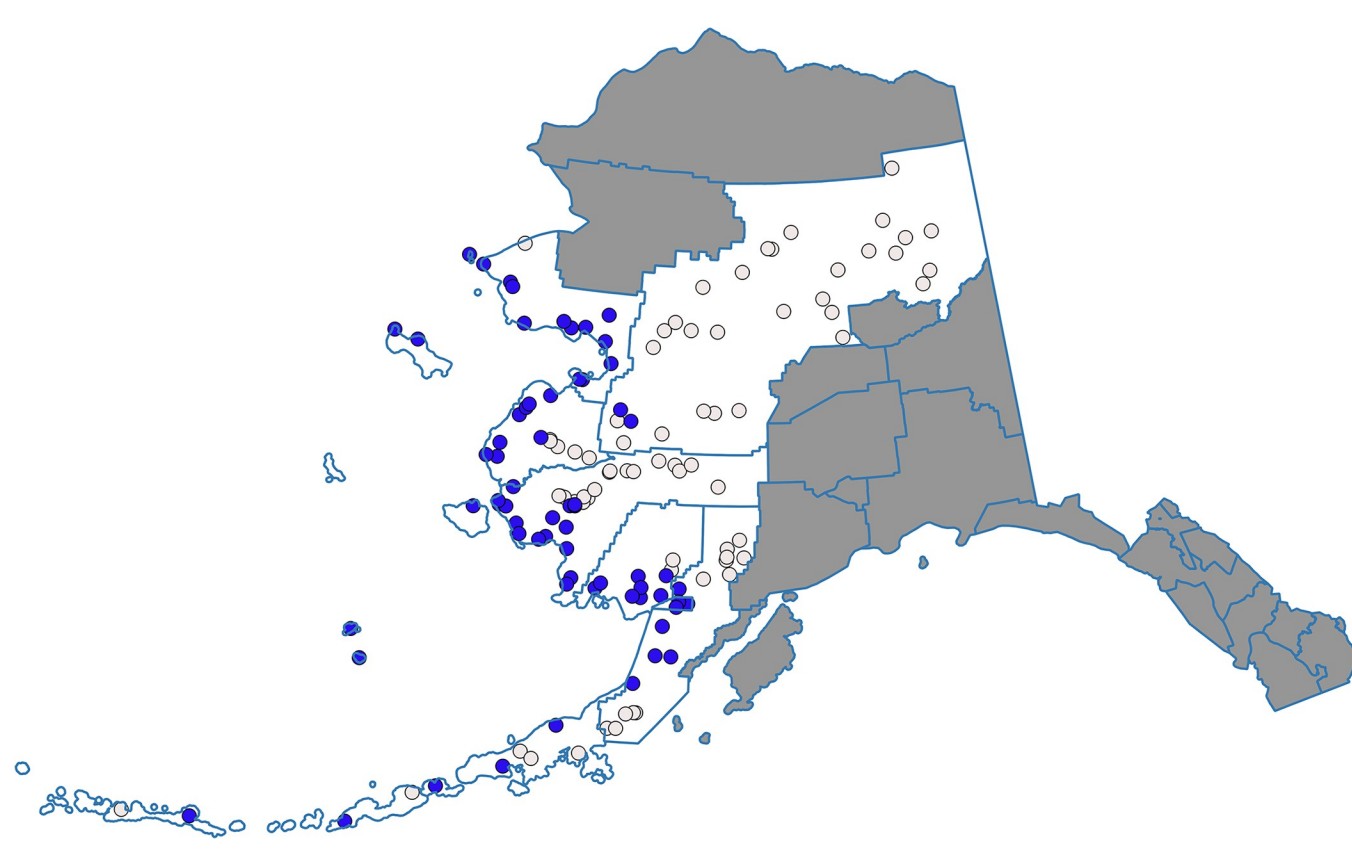

**Fig 1. Western Alaska community development quota communities.** Source: NOAA Fisheries [12].

(*Hippoglossus stenolepis*) and sablefish (*Anoplopoma fimbria*) when ITQs were implemented for these species. The remaining groundfish and crab fisheries were added in 1998. As fish prices increased over the years, so did the value of CDQ harvests. By 2016, harvests of groundfish, crab, and halibut by the CDQ groups had grown to 249,538 mt of seafood worth $120 million (NOAA 2018) [11].

The program gave CDQ organizations broad discretion over use of their fishing quotas, as long as proceeds were funneled to activities related to marine or coastal livelihoods. The fisheries involved were mainly large industrial fisheries, so did not provide direct fishing opportunities for small-scale community fisheries beyond a small halibut fishery in the Pribilof Islands. In this respect, the CDQ program design differs from the only other implementation of group rights to fisheries in North America–the Newfoundland-Labrador shrimp fishery—which awarded direct harvest rights to community fishing cooperatives [10,30]. Each CDQ organization had a board of directors with ultimate decision-making authority comprised of representatives selected by the constituent communities, with at least one board member representing each community served by that CDQ group. CDQ organizations embarked on a variety of different strategies to manage their quotas within program rules, as described in their respective annual reports [13]. Strategies ranged from leasing quota and using the lease revenue to upgrade small boat harbors and provide loans for fishing boats, to forming joint ventures and direct investments in the industrial fishery itself. Overall fishing quotas and harvest rules remain under federal government control, as recommended by the NPFMC and approved by the US Secretary of Commerce. Rather, the program allocates a set of non-transferable shares of the quota to groups of communities, as an alternative to ITQ systems commonly

implemented around the world. The CDQ program's unique approach to management of a commercial harvest quota share has captured the attention of a number of scholars over the years [24,31–33], as well as the National Research Council study completed early in the program's life. [15] No systematic evaluation of the longer term economic and social effects of the CDQ program has been conducted, however.

## Methods

Evaluation of the long-term effects of the CDQ program should consider the intended purposes: to provide western Alaska communities with the opportunity to participate and invest in fisheries in the region and to promote sustainable local economies providing economic and social benefits for residents. Evaluating potential economic and social benefits should also take into account that well-being in these largely Indigenous communities is a locally and culturally defined concept. The sponsors of the Arctic Human Development Report (AHDR) [34] recognized that the three dimensions, or domains, of well-being enumerated in the UN Human Development Report [35] and measured in the Human Development Index–education, health, material well-being–were insufficient to address well-being of Arctic peoples. The AHDR added cultural continuity, connection to the natural environment, and local autonomy to the three domains measured in the UN Human Development Report. The follow-on study to the AHDR–the Arctic Social Indicators (ASI) project [36,37]–defined a set of consensus indicators to measure the status of each of the six domains. These indicators, like the indicators in the UN Human Development Index, do not define well-being, but rather serve as a set of measurable social indicators that can be used to compare the status of a community, region, or nation relative to the status of another entity or to same entity over time. That makes the ASI indicators suitable for evaluating the effects of the CDQ program, with the understanding, as the ASI project explicitly acknowledged, that many aspects of the quality of life that are important to well-being and culture cannot be captured in a small set of quantifiable indicators.

One could argue that the CDQ program, by definition, represents an expansion of local autonomy, with its awarding of local control over a significant fraction of resource rights to one of the world's largest fisheries. While it would be appropriate to measure the main health indicator, infant mortality, data are unfortunately not systematically available at the local level, due to small populations and relatively low incidence generally throughout Alaska [38]. Point in time data on subsistence participation and harvest—primary indicators for the ASI contact with nature and cultural continuity domains—are available for many communities in Western Alaska [39]; however. subsistence data are unfortunately not systematically measured over time at the community level, as needed to test hypotheses about relative changes in subsistence associated with the CDQ program [40]. Consequently, we focused on indicators for the other three ASI domains of well-being, adding education and cultural continuity to several dimensions of material well-being, the primary stated objective of the CDQ program.

We applied a difference in differences approach to examine whether the CDQ program has increased well-being. The approach estimated how well-being outcomes changed for each of the indicators after the program was implemented, for residents of the 65 small, primarily Indigenous communities participating in the CDQ program, compared to changes in outcomes estimated over the same time period in adjacent otherwise similar communities not participating the program. To select comparison communities, we identified small, primarily Indigenous communities located in the same borough or Census Area (the Alaska equivalent of counties) as CDQ communities, but were ineligible to participate in the program. Ineligible communities included communities 50 or more miles from the Bering Sea, coastal communities along the Gulf of Alaska, and one community deemed to have established participation in

the Bering Sea industrial fishery. Eight of the nine Western Alaska boroughs and census areas that contain CDQ communities include one or more ineligible communities. Although CDQ and non-CDQ communities are culturally and economically similar and share much of the same colonial history, they nevertheless might have had somewhat different baseline levels of well-being before the CDQ program started. Consequently, we employed the difference in differences method to control for potential effects of uneven baseline levels of the measured indicators.

We examined 10 social and economic indicators that broadly represented the three AHDR domains of well-being in arctic Indigenous communities and were measured systematically at the community level. Specific indicators and hypothesis tests are enumerated below.

*Education*. One of the important ways the CDQ organizations contributed to communities is through the award of scholarships for higher education. There is no guarantee that the scholarship recipients would return to their communities after graduation; however, the opportunity to obtain a post-secondary scholarship could increase high school graduation rates. The potential for at least some college graduates to return to their community of origin motivates the following two hypotheses regarding educational attainment:

H1a: High school graduation rates increased more rapidly among American Indian and Alaska Native (AIAN) residents of CDQ communities than for residents of adjacent communities that were ineligible to participate in the program;

H1a: The likelihood that a community resident had a college degree increased more rapidly among AIAN residents of CDQ communities than for residents of adjacent communities that were ineligible to participate in the program.

*Cultural continuity*. The ASI report [36] specifically addressed Indigenous language retention as a primary indicator of cultural continuity. Although many communities in Western Alaska no longer speak the original Yupik, Inupiat, and Unangan languages, loss of language has been highly uneven. What is relevant for this study is not the level of loss, but the rate of loss since the program commenced. CDQ program objectives do not include language preservation; nevertheless, the program could affect language retention indirectly by providing support to local small-scale fishing livelihoods. These livelihoods provide opportunities for family and other community members to use their shared Indigenous language while working together in traditional coastal occupations that generate cash income along with subsistence foods [32].

H2: Although Indigenous language use may be declining throughout rural Alaska, the rate of decline has been slower among AIAN residents of CDQ communities than among those residing in adjacent ineligible communities.

*Material well-being*. We tested a variety of hypotheses about material well-being, including labor force participation, employment, earnings, and income.

H3a. The likelihood of year-round employment–working 40 weeks or more in a year–has increased among AIAN residents of CDQ communities compared to the change among residents of adjacent ineligible communities.

H3b. Annual per-capita earnings has increased among AIAN residents of CDQ communities compared to the change among residents of adjacent ineligible communities.

H3c. Annual per-capita income has increased among AIAN residents of CDQ communities compared to the change among residents of adjacent ineligible communities.

Changes in income inequality are difficult to measure in small communities. However, an important question to evaluate is whether the CDQ program was associated with changes at the lower end of the income scale.

H3d. The likelihood of having income below the poverty threshold has decreased among AIAN residents of CDQ communities compared to the change among residents of adjacent ineligible communities.

Because communities in Western Alaska are small–only two of the 65 communities contain more than 1,200 residents [41]—changes in community demographic composition could obscure changes in aggregate measures of well-being. For example, educational attainment has been increasing faster among women than men in rural Alaska [34], and poverty rates are higher among single-parent families with children [42]. One must be able to control for how potential demographic variation in each community might be associated with changes in well-being outcomes. To address that concern, we test the hypotheses about the local effects of the CDQ program using microdata on households and individuals to estimate the patterns of change in individual outcomes, controlling for demographic characteristics, rather than with aggregate community-level data.

## Data sources

The primary data sources for the study are individual and household records from the U.S. Census Long Form Surveys in 1990 and 2000, and American Community Surveys (ACS) from 2005 through 2016. The U.S. Census Bureau discontinued the Long Form Survey after the 2000 Census, replacing it with the ACS: an annual household survey with a similar questionnaire that the Bureau implemented in Alaska starting in 2005. The ACS and the former Long Form Survey represent the primary source of social and economic information on the U.S. population, and the only systematic recurring source of data on rural Alaska Indigenous peoples available at the community level. Access to individual and household records from the Census and ACS is protected by federal law [43], and release of information derived from individual and household records is highly restricted by the Census Bureau to prevent identification of individual information. We accessed the confidential records through a special arrangement with the Census Bureau Center for Economic Studies Research Data Center (RDC) program, approved 15 July 2015. The University of Alaska Anchorage Institutional Review Board approved Census Bureau ethics oversight of the project, which includes a waiver of informed consent, on 07 September 2012, reaffirmed on 05 October 2018. Confidential research data were accessed at the RDC under Census Bureau Supervision between 27 July 2015 and 30 January 2020.

Working at the RDC, we selected and matched household and person records for Alaska residents, adjusted data for different survey years for changes in questionnaire reporting, and combined data sets for years between 1990 and 2016 to generate a file including records over the 27-year period. The Census Long Form and ACS data do not represent a panel, and there is no way to identify individuals across years. Rather, we tested the hypotheses by comparing how outcomes for individuals and households with similar demographic characteristics had changed.

To estimate equations to test hypotheses about well-being indicators, we began by selecting records for all rural Alaska residents who indicated AIAN identify, either alone or in combination with other races. Rural Alaska is generally considered the area of the state that has no connection by road to urbanized areas, with a majority AIAN population and a distinctive economy [44]. We used the Census Bureau definition of rural Alaska: the 20 Census Areas and

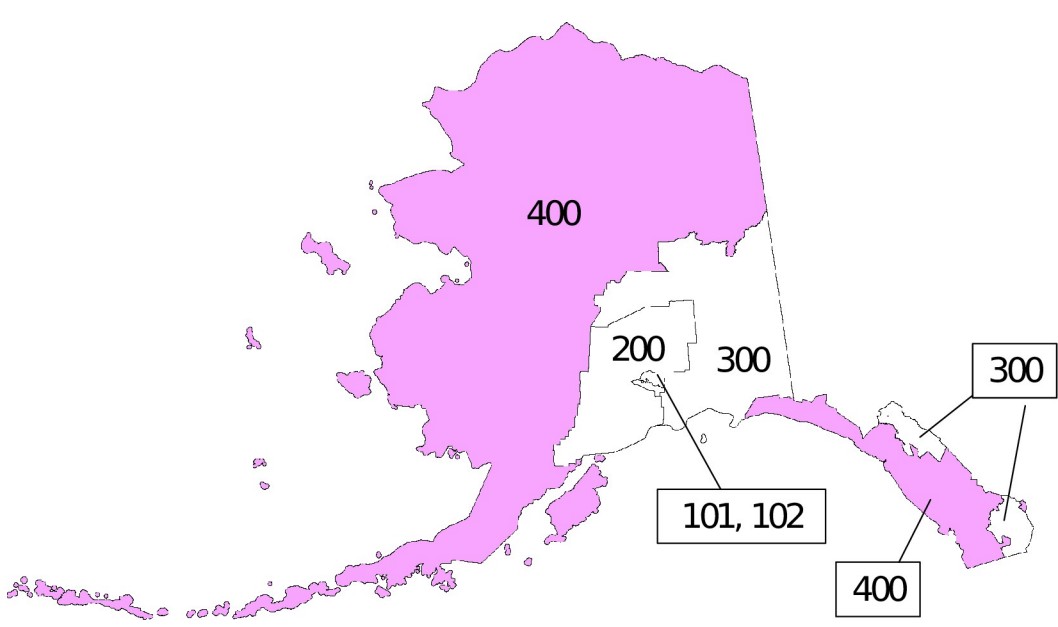

**Fig 2. 2010 census public use microdata areas in Alaska: Rural Alaska defined as PUMA 400 (Subsistence Alaska PUMA)** [45].

boroughs (county-level local governments) constituting Public Use Microdata Area (PUMA) 400, also known as the Subsistence Alaska Puma (Fig 2).

The final data set contained about 41,000 observations in the Census and ACS in 64 of the 65 CDQ-eligible communities; excluding one community (Ekuk) that was technically eligible had very few resident households. The data set also included about 31,000 person-records in 79 ineligible communities in the same 9 boroughs and census areas that contain CDQ communities, and 32,000 person-records in an additional 74 communities in the 9 other boroughs and census areas within the PUMA 400 region, for a total of 104,000 observations (numbers rounded to comply with Census Bureau disclosure policies). The definition of variables included in the data set, along with sample means and standard deviations are detailed S1 Table and S1 Appendix in S1 File.

## Statistical model

We hypothesize the following general relationship:

$$y_{ijt} = f(x_{ijt}, z_{jt}, t, k, Q)$$

where $y_{ijt}$ represents the social or economic status of individual or household $i$ residing in community $j$ located within Census Area or borough (county $k$), observed in year $t$; $x_{ijt}$ represents a vector of individual and household characteristics, $z_{jt}$ represents a vector of community-level demographic characteristics that could vary over time, and $Q$ represents whether or not the community was eligible to participate in the CDQ program. The specific form of the equations estimated may be written as follows:

$$y_{ijt} = \alpha x_{ijt} + \beta zjt + \gamma t + \varepsilon k + Q(\delta_1 t_1 - \delta_0 t_0),$$

where $\gamma t$ and $\varepsilon k$ are year and county fixed effects, respectively, time $t_1$ represents the time period after the CDQ program was underway (2000–2016), and $t_0$ represents the period before the program was implemented (1990).

To test hypotheses H3a, H3b, and H3c, we examine both individual and household outcomes, yielding 10 dependent variables to test the seven hypotheses. The individual survey records included in the equations vary somewhat for the different dependent variables, depending on which individuals were asked that question in the surveys. Language spoken in the home is a household variable and is not reported for residents of group quarters. The educational attainment and earnings variables are reported for individuals 16 years of age and older, including group quarters residents. Income is reported for individuals age 15 and older. We include group quarters residents in household weeks worked, earnings, and income by considering a group quarters resident to be equivalent to a single-person household. High school graduates, college graduates, 40 or more weeks worked (individual and household), and poverty status are binary variables. Annual earnings and income (individual and household) are continuous numeric variables. We censored the small number of negative reported individual earnings and income at zero.

Equations for binary variables were estimated as logistic regressions, using Stata, version 15 [46]. Equations for individual earnings and income and for household earnings were estimated as censored regressions. Because household income had very few censored observations, we used least squares regression to estimate household income. All equations were estimated with the Census person weight associated with the likelihood that the individual would have been sampled that year. Although the observations are for individuals, the values of the household dependent variables are identical for all individuals in each household. The values of poverty status are identical for each member of the family, which the Census Bureau defines as all related individuals living in the same household. We therefore adjusted the Wald statistics and other variance test statistics reported in the regression results for household language, household earnings, household income, and poverty status, by dividing by the average household (or family) size, and $z$ scores by the square root of household or family size.

## Results

Complete descriptions and summary statistics for all the variables included in the statistical analysis, along with full equation results are contained in S1 Appendix in S1 File. Many of the individual and household demographic characteristics showed large and significant effects on all well-being indicators. Here we summarize the estimated effects of the CDQ program, after controlling for all the individual, household, and community variables. Table 1 summarizes the estimated effects of participation in the CDQ program on Indigenous language retention,

**Table 1. Odds ratios for the estimated effect of the Western Alaska Community Development (CDQ) program on Indigenous language retention, educational attainment, employment, and poverty status.**

| Indicator | Odds ratio | | lower 95% | | upper 5% | |
|---|---|---|---|---|---|---|
| Indigenous language spoken at home | 104.5 | | 83.8 | | 130.4 | |
| High school degree | 88.5 | * | 76.8 | | 102.0 | |
| College degree | 94.1 | | 63.2 | | 140.1 | |
| Worked 40 weeks or more per year | 117.4 | ** | 100.3 | | 137.3 | |
| Any year-round worker in the household | 116.5 | | 95.8 | | 141.8 | |
| Income below poverty threshold | 81.7 | ** | 67.2 | | 99.4 | |

* p < 0.10.

** p < 0.05.

Source: Logistic regression results displayed in S2-S4 Tables in S1 File.

educational attainment, employment status, and poverty status. The estimated effects are shown as odds ratios from logistic regression equations.

## Language retention and educational attainment

Indigenous language use was much more prevalent in CDQ communities than in communities with comparable demographics and household composition in 1990 as well as in the period from 2000 onward (Census Bureau privacy rules prevent disclosure of the actual coefficients). The difference widened slightly after the CDQ program began–a 4.5 percent increase in the odds ratio between the two periods, shown in the first row of Table 1 –but the difference in the gap was not significantly different from zero. Equations including a time trend for CDQ communities relative to non-CDQ communities after 2000 suggest that the language disparity between CDQ and non-CDQ communities has been increasing by over two percent per year since 2000 (p = .01), which may in time lead to a significant difference between the two groups (Census Bureau privacy rules prevent disclosure of the actual coefficients).

After controlling for the effects of individual and household demographic characteristics, the CDQ program was associated with a slight reduction in the odds that the individual earned a high school degree, and a somewhat larger reduction in the odds of a college degree. In 2000 and later years, CDQ communities had a significantly lower prevalence of high school graduates (p < .01), compared to essentially no difference in 1990 before the program was implemented. However, the large standard error in 1990 makes the Wald test for the difference only marginally significant (p < .10) (Census Bureau privacy rules prevent disclosure of the actual coefficients). Individuals living in CDQ communities had a somewhat lower likelihood of possessing a college degree in 1990 as well as more recently, but the difference is not significant. However, when the differences in educational attainment are estimated excluding residents of the large hub communities, or excluding observations with Census Bureau imputed values for educational attainment, the apparent negative effect of the CDQ program becomes significant (p = .03 and p = .02, respectively) (Census Bureau privacy rules prevent disclosure of the actual coefficients).

## Year-round employment and poverty

Once all the strong demographic characteristics are taken into account, the results in Table 1 show that participation in the CDQ program was associated with a significant, positive effect on the likelihood the individual has year-round employment. The odds ratio that at least one individual in the household worked year-round is only slightly smaller than for the individual results. However, a slight downward trend observed after 2000 in the CDQ communities caused the CDQ effect overall to fall below the significance threshold. (Census Bureau privacy rules prevent disclosure of the actual coefficients). CDQ-eligible communities had fewer households with at least one adult working year-round in 1990 compared to non-CDQ communities, other things equal (p = .052). However, the CDQ communities caught up to the other communities, so that there was no difference between the two groups of communities after 2000 (p = .89) (Census Bureau privacy rules prevent disclosure of the actual coefficients).

The bottom row of Table 1 shows that after controlling for demographic factors, the CDQ program was associated with a significantly reduced odds that the income of an AIAN family member fell below the poverty threshold (p = .004). Poverty rates in the CDQ communities were higher in CDQ communities in 1990 relative to non-CDQ communities, but returned to levels of non-CDQ communities (probability of a difference = 0.47) after 2000, with no trends apparent thereafter (Census Bureau privacy rules prevent disclosure of the actual coefficients).

**Table 2. Effect of the Western Alaska Community Development (CDQ) program on earnings and income, estimated as percentage changes from baseline levels.**

| Indicator | Percentage change from baseline | | lower 95% | upper 5% |
|---|---|---|---|---|
| Annual individual earnings | 47.0 | ** | 5.9 | 104.0 |
| Annual individual income | 11.4 | ** | 1.2 | 22.6 |
| Annual household earnings | 38.7 | *** | 10.2 | 74.5 |
| Annual household income | 16.2 | *** | 7.4 | 25.7 |

** $p < 0.05$.

*** $p < 0.01$.

Source: Censored and ordinary least squares regression equations displayed in S3 and S4 Tables in S1 File.

## Earnings and income

Table 2 displays results for the natural logarithm of annual individual earnings and income, and per-capita household earnings and income of AIAN residents of CDQ and non-CDQ communities. The finding of significant positive association of the CDQ program with increased year-round employment would suggest a parallel finding with earnings and income. The results in Table 2 indicate that this was indeed the case. Once all the strong demographic characteristics are taken into account, the results show that participation in the CDQ program is associated with a significant, positive effect on individual earnings, and income. The magnitude of the effect on earnings–an average 47 percent increase—is striking given that the typically strongest predictor of earnings–educational attainment–is lower in the CDQ communities. The association of the CDQ program with income is also positive and significant, but much smaller in magnitude– 11 percent–compared to the very large effect on earnings. One may infer from this result that the increase in income in CDQ communities that occurred after 1990 consisted largely, if not wholly in the form of higher earned income.

The results for equations estimated for household earnings, income, shown in the last two rows of Table 2, largely mirror the results for the individual variables. Per-capita household earnings and incomes, like year-round employment, were significantly lower in CDQ communities in 1990 ($p < .05$ and $p < .01$, respectively), and poverty rates were higher ($p < .01$). Per-capita household earnings, however, were significantly higher in CDQ communities in 2000 and later ($p < .05$), while incomes and poverty rates rose to be equal to those in non-CDQ communities in and after 2000 (Census Bureau privacy rules prevent disclosure of the actual coefficients).

After controlling for demographic factors, per-capita household income was significantly lower in CDQ communities in 1990, especially in villages ($p < .001$), and the likelihood that family income was below the poverty threshold was significantly higher ($p = .004$) (equations not shown to protect confidentiality of responses). In both cases, income and poverty in the CDQ communities returned to levels close to those in non-CDQ communities after 2000 (probability of a difference = .19 for income, 0.47 for poverty), with no trends apparent after 2000 (complete equation results suppressed to protect confidentiality of respondents).

## Discussion

Measuring the effects of the CDQ program on social and economic status of the rural Alaska AIAN population requires controlling for a broad array of demographic characteristics, which can vary substantially across communities and over time and obscure community differences in outcomes. The ability to access Census records enables a much more precise and robust

accounting for the influence of these characteristics, giving one the ability to examine how outcomes for individuals and households living in CDQ communities with a given set of demographic characteristics compare to outcomes for similar households living in nearby communities that were ineligible for the program. "Nearby" ineligible communities were defined as similarly small, Indigenous communities located in the same county as the CDQ community.

The results estimated for a large sample of AIAN individuals age 16 and older (>65,000) over a 27-year period showed significant improvements in material well-being outcomes for residents of CDQ-eligible communities, relative to the respective changes in outcomes for demographically similar residents of the CDQ-ineligible communities over the same time period. CDQ residents on average experienced relatively large positive gains in earnings, along with a higher likelihood full-time employment, increased income, and a reduction in poverty compared to similar residents of non-CDQ communities. Although the CDQ program did alleviate poverty, CDQ communities started at a disadvantage relative to the comparison communities, and the magnitude of poverty alleviation has only been sufficient to reduce poverty rates in these communities to match poverty rates in non-CDQ communities. The results showed no significant effect of the program on the use of Indigenous language at home, suggesting that the improvement in economic status was not accomplished at the expense of a loss of Indigenous heritage.

Although the difference in differences method controls for most social ane economic changes, it cannot control for all changes that might systematically affect the CDQ communities differently to the adjacent comparison communities. Of particular concern are potential effects of the small-scale non-CDQ commercial fisheries. State fishery participation and landings data show that gross earnings and participation in commercial fisheries in small Western Alaska communities have declined steadily since 1990 [47]. The rate of decline, however, is similar in CDQ fisheries and the non-CDQ comparison communities (Fig 3). The comparison communities also participate in commercial fisheries, but the CDQ communities' greater dependence on commercial fishing would suggest that they were bucking economic headwinds, yet still improved their relative performance.

Although material well-being improved, there was no evidence that educational attainment increased in the CDQ communities. In fact, the proportion of residents with a high-school degree actually increased slightly more in the non-CDQ communities than in the CDQ

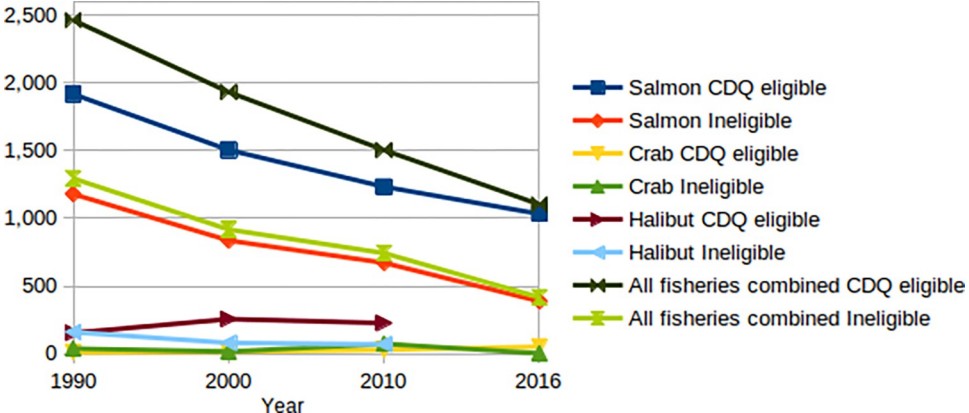

**Fig 3. Number of commercial fishery permit-holders making landings by fishery group for CDQ and non-CDQ comparison communities, 1990–2016.** Data for all fisheries combined exclude halibut, which are not reported. Source: Alaska Commercial Fisheries Entry Commission [47].

communities. The proportion with college degrees was unchanged relative to the control communities, despite the award of college scholarships offered by most of the CDQ organizations. Although CDQ groups offered an average 660 scholarships for post-secondary education [13], many of these scholarships were used for vocational training that would not lead to a college degree. The lack of relative improvement in education has a number of potential explanations. At the high school level, school district boundaries for most CDQ communities are drawn at the county level, making it unlikely that a CDQ organization could easily influence differential educational resources or policies for its constituent communities relative to the ineligible communities. Many ANSCA corporations and tribal governments offer scholarships, too, and it is possible that those organizations favor non-CDQ communities to offset the increased resources available to CDQ communities. CDQ economic investments were directed by statute to favor local fisheries, an industry for which formal education is generally not a requirement. Without more employment opportunities that required a college degree, CDQ community residents leaving the community to attend college would have little incentive to return.

The findings discussed are limited by the availability of data that can be measured and compared over time: a limitation inherent to quantitative evaluations generally. Systematic availability of community-level data over time limited the analysis to indicators measured in Census data. A significant challenge is that neither the Census Bureau nor any other source provides systematic measurements of subsistence participation and harvests. Although the CDQ program covers 65 communities, they are represented by 6 separate entities, each of which has implemented a different set of policies to meet program requirements. This diversity, when coupled with Census privacy rules, leaves insufficient statistical power to distinguish statistical differences among different CDQ groups related to decision-making processes or development strategies, let alone differences specific to communities [34].

## Conclusion

The statistical analysis appears to support the claims of proponents of the CDQ program that awarding group rights to fisheries, in the form of organizations representing coalitions of communities, can improve material well-being without contributing to loss of culture or identity, as measured by Indigenous language retention. Although there do not appear to be any regional economic trends that might be affecting the CDQ communities differently from adjacent ineligible communities, the data available for this study are insufficient to rule out that possibility. This study was limited to statistical analysis. Additional community level qualitative research is needed to evaluate program social and economic benefits more broadly, and understand how the program benefits are realized and distributed locally within communities.

## Supporting information

**S1 File. PLoS appendix.** S1 Appendix. Detailed Statistical Results.
(DOCX)

## Acknowledgments

Rachelle Wang-Cendejas provided assistance with data management. Logistical support was provided by the U.S. Census Bureau Center for Economic Studies and the California Census Research Data Center at the University of Southern California.

## Author Contributions

**Conceptualization:** Matthew Berman.

**Data curation:** Matthew Berman.

**Formal analysis:** Matthew Berman.

**Funding acquisition:** Matthew Berman.

**Investigation:** Matthew Berman.

**Methodology:** Matthew Berman.

**Project administration:** Matthew Berman.

**Supervision:** Matthew Berman.

**Validation:** Matthew Berman.

**Writing – original draft:** Matthew Berman.

**Writing – review & editing:** Matthew Berman.

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
