## [Decision Letter · Decision Letter 0]

15 Jul 2024

PONE-D-24-19040Long-term effects of group rights to fisheries: Evaluating the Western Alaska Community Development Quota programPLOS ONE

Dear Dr. Berman,

Thank you for submitting your manuscript to PLOS ONE. I have received two reviews of the manuscript and both reviewers conclude this is a valuable contribution, although they both also have suggestions that I think will improve the manuscript, specifically issues such as carefully defining and using acronyms for audiences not familiar with western Alaska and otherwise providing context. Also the suggestion of providing a map with community locations strikes me as helpful for broader understanding of the study, and another point that resonated with me was providing linkages to important goals such as preserving Indigenous languages to outcomes of the CDQ approach. Given these constructive suggestions, I invite you to submit a revised version of the manuscript that addresses the points raised during the review process.

A marked-up copy of your manuscript that highlights changes made to the original version. You should upload this as a separate file labeled 'Revised Manuscript with Track Changes'.An unmarked version of your revised paper without tracked changes. You should upload this as a separate file labeled 'Manuscript'.If applicable, we recommend that you deposit your laboratory protocols in protocols.io to enhance the reproducibility of your results. Protocols.io assigns your protocol its own identifier (DOI) so that it can be cited independently in the future. For instructions see: https://journals.plos.org/plosone/s/submission-guidelines#loc-laboratory-protocols. Additionally, PLOS ONE offers an option for publishing peer-reviewed Lab Protocol articles, which describe protocols hosted on protocols.io. Read more information on sharing protocols at https://plos.org/protocols?utm_medium=editorial-email&utm_source=authorletters&utm_campaign=protocols.

We look forward to receiving your revised manuscript.

Kind regards,

Lee W Cooper, Ph.D.

Section Editor

PLOS ONE

 [National Science Foundation, award #1216399].  

3. In the online submission form you indicate that your data is not available for proprietary reasons and have provided a contact point for accessing this data. Please note that your current contact point is a co-author on this manuscript. According to our Data Policy, the contact point must not be an author on the manuscript and must be an institutional contact, ideally not an individual. Please revise your data statement to a non-author institutional point of contact, such as a data access or ethics committee, and send this to us via return email. Please also include contact information for the third party organization, and please include the full citation of where the data can be found.

Additional Editor Comments (if provided):

Reviewers' comments:

Reviewer's Responses to Questions

**Comments to the Author**

1. Is the manuscript technically sound, and do the data support the conclusions?

Reviewer #1: Yes

Reviewer #2: Yes

2. Has the statistical analysis been performed appropriately and rigorously? 

Reviewer #1: Yes

Reviewer #2: I Don't Know

3. Have the authors made all data underlying the findings in their manuscript fully available?

Reviewer #1: No

Reviewer #2: Yes

4. Is the manuscript presented in an intelligible fashion and written in standard English?

Reviewer #1: Yes

Reviewer #2: Yes

5. Review Comments to the Author

Reviewer #1: This manuscript, “Long-term effects of group rights to fisheries: Evaluating the Western Alaska Community Development Quota program”, is a very well written and researched investigation of the effect of the Western Alaska Community Development Quota program on aspects of well-being on participating communities. This is an important piece of work for the fisheries literature and fisheries management, as other regions are working on pursuing a similar quota program for small-scale fisheries. The statistical approach, differences-in-differences, is very appropriate for the data and hypotheses.

Minor comments:

1. This appears to be a sole-authored manuscript. It would seem more appropriate to use the singular than plural (e.g., paragraph starting line 70).

2. This manuscript contains quite a few acronyms that are hard to follow for someone unfamiliar with them. I would suggest spelling out the acronyms that are not used frequently (e.g., BSAI) to improve readability.

3. Methods, first paragraph. A nice addition to the manuscript would be a table that summarizes the six consensus indicators, highlights the three that were used, and the specific metrics and data sources / variables that were used in analyses.

4. Sentence line 171-173: awkward – perhaps a word is missing or superfluous. Check the sentence.

5. Line 207: Presumably this should be hypothesis H1b

6. Table 1 and 2: it would be good to add the hypotheses to the tables so that it is easier to link the results back to the hypotheses.

7. Table 2: check the legend for significance. It should different numbers of *s than the table

Reviewer #2: This is a well-written and structured article, but I think it can be improved upon in a few ways. Most of my suggestions have to do with appropriate framing and further unpacking of results.

Firstly, I think the paper should explicitly mention that well-being, especially Indigenous well-being, is multidimensional and a community- and culturally- defined concept.

The paper mentions drawing on a definition of well-being provided by the Arctic Human Development Report without stating what that definition is, or that it may not be universally accepted or adopted by a region as diverse as western Alaska. It’s important also to mention that this analysis is limited in scope to social and economic change or benefits that can be systematically measured across communities and thus many important dimensions of well-being and culture remain unaccounted for in the study. This will help to frame or make more acceptable to the reader the selection of indicators as described later on in the paper, for example, the single indicator of Indigenous language retention for cultural continuity.

Language retention indicator: This single indicator is somewhat problematic as a measure of cultural continuity given the high level of Indigenous language loss that has occurred across Alaska. Many studies suggest that the continuation of traditional hunting and fishing practices and related food practices is vital to Indigenous well-being and a central facet of cultural continuity. The paper states that they don’t have access to good data to bring subsistence practices and foods into the analysis but I would encourage the authors to consider adding at the very least a paragraph in the ‘BS fisheries and communities’ section that describes the mixed-economies of western Alaska and the high reliance and value placed on subsistence practices and foods.

As currently written it’s unclear how the CDQ program benefits or supports language retention or immersion programs? The authors provide two examples (lines 212-214) but they seem tenuous or perhaps assumed and I’m not sure if that was the intention. To my knowledge, it is primarily Alaska Native Corporations and Tribes that support/lead the efforts to restore Indigenous languages in western AK. If CDQs are involved in language restoration initiatives, providing a few examples to better make the linkage would be helpful.

Figures were not included in the PDF to review so this may not be an issue but

a map of the CDQ region/regions would be really helpful to the reader, especially if the map included where communities are located so the reader can see communities included/excluded from the program. The paper references nearby ineligible communities in the same county, but not a lot of detail about this subset; a map or a list of CDQ and non-CDQ communities included in the analysis may be helpful.

Individual earnings and income:

This paper mentions that larger economic changes were controlled for in the analysis but it was unclear if and how engagement in non-CDQ commercial fisheries may have shaped results related to earnings and income in CDQ regions. Is there a way to tease these factors apart? For example, I was thinking about the role of state managed salmon, crab, or other (non-CDQ) fisheries and the economic engine they serve in many coastal (CDQ) communities. Oftentimes these communities reap greater benefits than those further upriver or inland because of greater access to commercial fisheries, processing, infrastructure, and ancillary services. These wouldn’t necessarily be CDQ related benefits. Is there a way to control for participation in local fisheries as a contributor to large income benefits or to better account for it in the results section?

Finally, most Tribes and Native corporations offer scholarships that would be available to non-CDQ students so that may help to explain the lack of difference in educational outcomes between the two sets of communities if focusing specifically on Alaska Native households.

What surprises me most about these results is the data related to changes in poverty levels. Findings suggest that poverty rates in CDQ communities have dropped since 1990 but they remain equal to rates in non-CDQ communities since 2000. Is that correct? This is surprising given that one of the main missions of the CDQ program is to alleviate poverty.

Other comments:

Line 40 – missing word at end of sentence?

Lines 45-47 - The authors should consider reviewing Paul Foley’s work on community rights in Canada. He has many recent papers that should be of interest but I’ve listed two here from 2015 that are especially relevant:

• Governing enclosure for coastal communities: Social embeddedness in a Canadian shrimp fishery (2015), Marine Policy

• Making Space for Community Use Rights: Insights From “Community Economies” in Newfoundland and Labrador (2015), Society and Natural Resources

Lines 153-4 – should cite Courtney Lyons work here.

• Alaska's community development quota program: A complex institution affecting rural communities in disparate ways (2019), Marine Policy

In the Intro or Methods section it should be made clear that the CDQ program doesn’t provide actual fishing opportunity for residents of the region outside of the halibut fishery.

6. PLOS authors have the option to publish the peer review history of their article (what does this mean?). If published, this will include your full peer review and any attached files.

Reviewer #1: No

Reviewer #2: No

---

## [Author Response · Author response to Decision Letter 0]

8 Oct 2024

Response to editor comments included in the cover letter. Responses to review comments are included in a document uploaded with the manuscript.

---

## [Editor Report · Decision Letter 1]

11 Oct 2024

Long-term effects of group rights to fisheries: Evaluating the Western Alaska Community Development Quota program

PONE-D-24-19040R1

Dear Matthew,

Thank you for returning your revised manuscript and thoroughly addressing the recommendations made by the two reviewers. I'm pleased to inform you that your manuscript has been judged scientifically suitable for publication and will be formally accepted for publication once it meets any outstanding technical requirements identified by the Editorial Office.

Kind regards,

Lee W Cooper, Ph.D.

Section Editor

PLOS ONE

---

## [Editor Report · Acceptance letter]

24 Oct 2024

PONE-D-24-19040R1 

PLOS ONE

Dear Dr. Berman, 

I'm pleased to inform you that your manuscript has been deemed suitable for publication in PLOS ONE. Congratulations! Your manuscript is now being handed over to our production team.

Kind regards, 

on behalf of

Dr. Lee W Cooper 

Section Editor

PLOS ONE